# Pixels to Graphs by Associative Embedding

**Alejandro Newell**       **Jia Deng**
Computer Science and Engineering
University of Michigan, Ann Arbor
`{alnewell, jiadeng}@umich.edu`

## Abstract

Graphs are a useful abstraction of image content. Not only can graphs represent details about individual objects in a scene but they can capture the interactions between pairs of objects. We present a method for training a convolutional neural network such that it takes in an input image and produces a full graph definition. This is done end-to-end in a single stage with the use of associative embeddings. The network learns to simultaneously identify all of the elements that make up a graph and piece them together. We benchmark on the Visual Genome dataset, and demonstrate state-of-the-art performance on the challenging task of scene graph generation.

## 1   Introduction

Extracting semantics from images is one of the main goals of computer vision. Recent years have seen rapid progress in the classification and localization of objects [7, 24, 10]. But a bag of labeled and localized objects is an impoverished representation of image semantics: it tells us what and where the objects are ("person" and "car"), but does not tell us about their relations and interactions ("person next to car"). A necessary step is thus to not only detect objects but to identify the relations between them. An explicit representation of these semantics is referred to as a *scene graph* [12] where we represent objects grounded in the scene as vertices and the relationships between them as edges.

End-to-end training of convolutional networks has proven to be a highly effective strategy for image understanding tasks. It is therefore natural to ask whether the same strategy would be viable for predicting graphs from pixels. Existing approaches, however, tend to break the problem down into more manageable steps. For example, one might run an object detection system to propose all of the objects in the scene, then isolate individual pairs of objects to identify the relationships between them [18]. This breakdown often restricts the visual features used in later steps and limits reasoning over the full graph and over the full contents of the image.

We propose a novel approach to this problem, where we train a network to define a complete graph from a raw input image. The proposed supervision allows a network to better account for the full image context while making predictions, meaning that the network reasons jointly over the entire scene graph rather than focusing on pairs of objects in isolation. Furthermore, there is no explicit reliance on external systems such as Region Proposal Networks (RPN) [24] that provide an initial pool of object detections.

To do this, we treat all graph elements—both vertices and edges—as visual entities to be detected as in a standard object detection pipeline. Specifically, a vertex is an instance of an object ("person"), and an edge is an instance of an object-object relation ("person next to car"). Just as visual patterns in an image allow us to distinguish between objects, there are properties of the image that allow us to see relationships. We train the network to pick up on these properties and point out where objects and relationships are likely to exist in the image space.

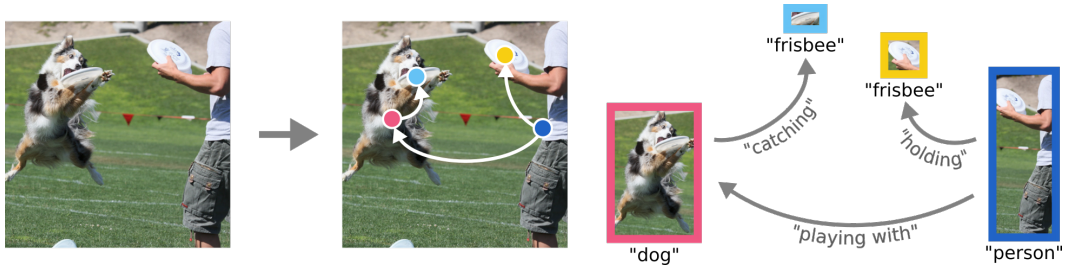

Figure 1: Scene graphs are defined by the objects in an image (vertices) and their interactions (edges). The ability to express information about the connections between objects make scene graphs a useful representation for many computer vision tasks including captioning and visual question answering.

What distinguishes this work from established detection approaches [24] is the need to represent connections between detections. Traditionally, a network takes an image, identifies the items of interest, and outputs a pile of independent objects. A given detection does not tell us anything about the others. But now, if the network produces a pool of objects ("car", "person", "dog", "tree", etc), and also identifies a relationship such as "in front of" we need to define which of the detected objects is in front of which. Since we do not know which objects will be found in a given image ahead of time, the network needs to somehow refer to its own outputs.

We draw inspiration from associative embeddings [20] to solve this problem. Originally proposed for detection and grouping in the context of multiperson pose estimation, associative embeddings provide the necessary flexibility in the network's output space. For pose estimation, the idea is to predict an embedding vector for each detected body joint such that detections with similar embeddings can be grouped to form an individual person. But in its original formulation, the embeddings are too restrictive, the network can only define clusters of nodes, and for a scene graph, we need to express arbitrary edges between pairs of nodes.

To address this, associative embeddings must be used in a substantially different manner. That is, rather than having nodes output a shared embedding to refer to clusters and groups, we instead have each node define its own unique embedding. Given a set of detected objects, the network outputs a different embedding for each object. Now, each edge can refer to the source and destination nodes by correctly producing their embeddings. Once the network is trained it is straightforward to match the embeddings from detected edges to each vertex and construct a final graph.

There is one further issue that we address in this work: how to deal with detections grounded at the same location in the image. Frequently in graph prediction, multiple vertices or edges may appear in the same place. Supervision of this is difficult as training a network traditionally requires telling it exactly what appears and where. With an unordered set of overlapping detections there may not be a direct mapping to explicitly lay this out. Consider a set of object relations grounded at the same pixel location. Assume the network has some fixed output space consisting of discrete "slots" in which detections can appear. It is unclear how to define a mapping so that the network has a consistent rule for organizing its relation predictions into these slots. We address this problem by not enforcing any explicit mapping at all, and instead provide supervision such that it does not matter how the network chooses to fill its output, a correct loss can still be applied.

Our contributions are a novel use of associative embeddings for connecting the vertices and edges of a graph, and a technique for supervising an unordered set of network outputs. Together these form the building blocks of our system for direct graph prediction from pixels. We apply our method to the task of generating a semantic graph of objects and relations and test on the Visual Genome dataset [14]. We achieve state-of-the-art results improving performance over prior work by nearly a factor of three on the most difficult task setting.

## 2   Related Work

**Relationship detection:** There are many ways to frame the task of identifying objects and the relationships between them. This includes localization from referential expressions [11], detection of human-object interactions [3], or the more general tasks of visual relationship detection (VRD) [18] and scene graph generation [12]. In all of these settings, the aim is to correctly determine the

relationships between pairs of objects and ground this in the image with accurate object bounding boxes.

Visual relationship detection has drawn much recent attention [18, 28, 27, 2, 17, 19, 22, 23]. The open-ended and challenging nature of the task lends itself to a variety of diverse approaches and solutions. For example: incorporating vision and language when reasoning over a pair of objects [18]; using message-passing RNNs to process a set of proposed object boxes [26]; predicting over triplets of bounding boxes that corresponding to proposals for a *subject, phrase,* and *object* [15]; using reinforcement learning to sequentially evaluate on pairs of object proposals and determine their relationships [16]; comparing the visual features and relative spatial positions of pairs of boxes [4]; learning to project proposed objects into a vector space such that the difference between two object vectors is informative of the relationship between them [27].

Most of these approaches rely on generated bounding boxes from a Region Proposal Network (RPN) [24]. Our method does not require proposed boxes and can produce detections directly from the image. However proposals can be incorporated as additional input to improve performance. Furthermore, many methods process pairs of objects in isolation whereas we train a network to process the whole image and produce all object and relationship detections at once.

**Associative Embedding:** Vector embeddings are used in a variety of contexts. For example, to measure the similarity between pairs of images [6, 25], or to map visual and text features to a shared vector space [5, 8, 13]. Recent work uses vector embeddings to group together body joints for multiperson pose estimation [20]. These are referred to as associative embeddings since supervision does not require the network to output a particular vector value, and instead uses the distances between pairs of embeddings to calculate a loss. What is important is not the exact value of the vector but how it relates to the other embeddings produced by the network.

More specifically, in [20] a network is trained to detect body joints of the various people in an image. In addition, it must produce a vector embedding for each of its detections. The embedding is used to identify which person a particular joint belongs to. This is done by ensuring that all joints that belong to a single individual produce the same output embedding, and that the embeddings across individuals are sufficiently different to separate detections out into discrete groups. In a certain sense, this approach does define a graph, but the graph is restricted in that it can only represent clusters of nodes. For the purposes of our work, we take a different perspective on the associative embedding loss in order to express any arbitrary graph as defined by a set of vertices and directed edges. There are other ways that embeddings could be applied to solve this problem, but our approach depends on our specific formulation where we treat edges as elements of the image to be detected which is not obvious given the prior use of associative embeddings for pose.

## 3 Pixels → Graph

Our goal is to construct a graph from a set of pixels. In particular, we want to construct a graph grounded in the space of these pixels. Meaning that in addition to identifying vertices of the graph, we want to know their precise locations. A vertex in this case can refer to any object of interest in the scene including people, cars, clothing, and buildings. The relationships between these objects is then captured by the edges of the graph. These relationships may include verbs (*eating, riding*), spatial relations (*on the left of, behind*), and comparisons (*smaller than, same color as*).

More formally we consider a directed graph $G = (V, E)$. A given vertex $v_i \in V$ is grounded at a location $(x_i, y_i)$ and defined by its class and bounding box. Each edge $e \in E$ takes the form $e_i = (v_s, v_t, r_i)$ defining a relationship of type $r_i$ from $v_s$ to $v_t$. We train a network to explicitly define $V$ and $E$. This training is done end-to-end on a single network, allowing the network to reason fully over the image and all possible components of the graph when making its predictions.

While production of the graph occurs all at once, it helps to think of the process in two main steps: detecting individual elements of the graph, and connecting these elements together. For the first step, the network indicates where vertices and edges are likely to exist and predicts the properties of these detections. For the second, we determine which two vertices are connected by a detected edge. We describe these two steps in detail in the following subsections.

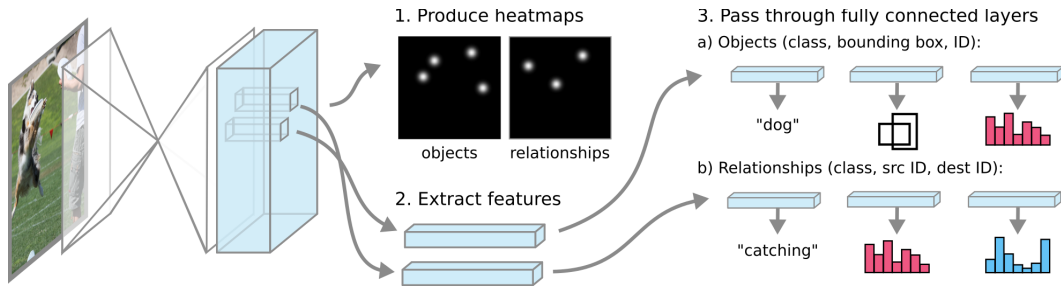

Figure 2: Full pipeline for object and relationship detection. A network is trained to produce two heatmaps that activate at the predicted locations of objects and relationships. Feature vectors are extracted from the pixel locations of top activations and fed through fully connected networks to predict object and relationship properties. Embeddings produced at this step serve as IDs allowing detections to refer to each other.

## 3.1 Detecting graph elements

First, the network must find all of the vertices and edges that make up a graph. Each graph element is grounded at a pixel location which the network must identify. In a scene graph where vertices correspond to object detections, the center of the object bounding box will serve as the grounding location. We ground edges at the midpoint of the source and target vertices: $(\lfloor \frac{x_s + x_t}{2} \rfloor, \lfloor \frac{y_s + y_t}{2} \rfloor)$. With this grounding in mind, we can detect individual elements by using a network that produces per-pixel features at a high output resolution. The feature vector at a pixel determines if an edge or vertex is present at that location, and if so is used to predict the properties of that element.

A convolutional neural network is used to process the image and produce a feature tensor of size $h$ x $w$ x $f$. All information necessary to define a vertex or edge is thus encoded at particular pixel in a feature vector of length $f$. Note that even at a high output resolution, multiple graph elements may be grounded at the same location. The following discussion assumes up to one vertex and edge can exist at a given pixel, and we elaborate on how we accommodate multiple detections in Section 3.3.

We use a stacked hourglass network [21] to process an image and produce the output feature tensor. While our method has no strict dependence on network architecture, there are some properties that are important for this task. The hourglass design combines global and local information to reason over the full image and produce high quality per-pixel predictions. This is originally done for human pose prediction which requires global reasoning over the structure of the body, but also precise localization of individual joints. Similar logic applies to scene graphs where the context of the whole scene must be taken into account, but we wish to preserve the local information of individual elements.

An important design choice here is the output resolution of the network. It does not have to match the full input resolution, but there are a few details worth considering. First, it is possible for elements to be grounded at the exact same pixel. The lower the output resolution, the higher the probability of overlapping detections. Our approach allows this, but the fewer overlapping detections, the better. All information necessary to define these elements must be encoded into a single feature vector of length $f$ which gets more difficult to do as more elements occupy a given location. Another detail is that increasing the output resolution aids in performing better localization.

To predict the presence of graph elements we take the final feature tensor and apply a 1x1 convolution and sigmoid activation to produce two heatmaps (one for vertices and another for edges). Each heatmap indicates the likelihood that a vertex or edge exists at a given pixel. Supervision is a binary cross-entropy loss on the heatmap activations, and we threshold on the result to produce a candidate set of detections.

Next, for each of these detections we must predict their properties such as their class label. We extract the feature vector from the corresponding location of a detection, and use the vector as input to a set of fully connected networks. A separate network is used for each property we wish to predict, and each consists of a single hidden layer with $f$ nodes. This is illustrated above in Figure 2. During training we use the ground truth locations of vertices and edges to extract features. A softmax loss is used to supervise labels like object class and relationship predicate. And to predict bounding box information we use anchor boxes and regress offsets based on the approach in Faster-RCNN [24].

In summary, the detection pipeline works as follows: We pass the image through a network to produce a set of per-pixel features. These features are first used to produce heatmaps identifying vertex and edge locations. Individual feature vectors are extracted from the top heatmap locations to predict the appropriate vertex and edge properties. The final result is a pool of vertex and edge detections that together will compose the graph.

## 3.2 Connecting elements with associative embeddings

Next, the various pieces of the graph need to be put together. This is made possible by training the network to produce additional outputs in the same step as the class and bounding box prediction. For every vertex, the network produces a unique identifier in the form of a vector embedding, and for every edge, it must produce the corresponding embeddings to refer to its source and destination vertices. The network must learn to ensure that embeddings are different across different vertices, and that all embeddings that refer to a single vertex are the same.

These embeddings are critical for explicitly laying out the definition of a graph. For instance, while it is helpful that edge detections are grounded at the midpoint of two vertices, this ultimately does not address a couple of critical details for correctly constructing the graph. The midpoint does not indicate which vertex serves as the source and which serves as the destination, nor does it disambiguate between pairs of vertices that happen to share the same midpoint.

To train the network to produce a coherent set of embeddings we build off of the loss penalty used in [20]. During training, we have a ground truth set of annotations defining the unique objects in the scene and the edges between these objects. This allows us to enforce two penalties: that an edge points to a vertex by matching its output embedding as closely as possible, and that the embedding vectors produced for each vertex are sufficiently different. We think of the first as "pulling together" all references to a single vertex, and the second as "pushing apart" the references to different individual vertices.

We consider an embedding $h_i \in R^d$ produced for a vertex $v_i \in V$. All edges that connect to this vertex produce a set of embeddings $h'_{ik}, k = 1, ..., K_i$ where $K_i$ is the total number of references to that vertex. Given an image with $n$ objects the loss to "pull together" these embeddings is:

$$L_{pull} = \frac{1}{\sum_{i=1}^{n} K_i} \sum_{i=1}^{n} \sum_{k=1}^{K_i} (h_i - h'_{ik})^2$$

To "push apart" embeddings across different vertices we first used the penalty described in [20], but experienced difficulty with convergence. We tested alternatives and the most reliable loss was a margin-based penalty similar to [9]:

$$L_{push} = \sum_{i=1}^{n-1} \sum_{j=i+1}^{n} max(0, m - ||h_i - h_j||)$$

Intuitively, $L_{push}$ is at its highest the closer $h_i$ and $h_j$ are to each other. The penalty drops off sharply as the distance between $h_i$ and $h_j$ grows, eventually hitting zero once the distance is greater than a given margin $m$. On the flip side, for some edge connected to a vertex $v_i$, the loss $L_{pull}$ will quickly grow the further its reference embedding $h'_i$ is from $h_i$.

The two penalties are weighted equally leaving a final associative embedding loss of $L_{pull} + L_{push}$. In this work, we use $m = 8$ and $d = 8$. Convergence of the network improves greatly after increasing the dimension $d$ of tags up from 1 as used in [20].

Once the network is trained with this loss, full construction of the graph can be performed with a trivial postprocessing step. The network produces a pool of vertex and edge detections. For every edge, we look at the source and destination embeddings and match them to the closest embedding amongst the detected vertices. Multiple edges may have the same source and target vertices, $v_s$ and $v_t$, and it is also possible for $v_s$ to equal $v_t$.

### 3.3 Support for overlapping detections

In scene graphs, there are going to be many cases where multiple vertices or multiple edges will be grounded at the same pixel location. For example, it is common to see two distinct relationships between a single pair of objects: *person wearing shirt — shirt on person*. The detection pipeline must therefore be extended to support multiple detections at the same pixel.

One way of dealing with this is to define an extra axis that allows for discrete separation of detections at a given $x, y$ location. For example, one could split up objects along a third spatial dimension assuming the z-axis were annotated, or perhaps separate them by bounding box anchors. In either of these cases there is a visual cue guiding the network so that it can learn a consistent rule for assigning new detections to a correct slot in the third dimension. Unfortunately this idea cannot be applied as easily to relationship detections. It is unclear how to define a third axis such that there is a reliable and consistent bin assignment for each relationship.

In our approach, we still separate detections out into several discrete bins, but address the issue of assignment by not enforcing any specific assignment at all. This means that for a given detection we strictly supervise the $x, y$ location in which it is to appear, but allow it to show up in one of several "slots". We have no way of knowing ahead of time in which slot it will be placed by the network, so this means an extra step must be taken at training time to identify where we think the network has placed its predictions and then enforce the loss at those slots.

We define $s_o$ and $s_r$ to be the number of slots available for objects and relationships respectively. We modify the network pipeline so that instead of producing predictions for a single object and relationship at a pixel, a feature vector is used to produce predictions for a set of $s_o$ objects and $s_r$ relationships. That is, given a feature vector $f$ from a single pixel, the network will for example output $s_o$ object class labels, $s_o$ bounding box predictions, and $s_o$ embeddings. This is done with separate fully connected layers predicting the various object and relationship properties for each available slot. No weights are shared amongst these layers. Furthermore, we add an additional output to serve as a score indicating whether or not a detection exists at each slot.

During training, we have some number of ground truth objects, between 1 and $s_o$, grounded at a particular pixel. We do not know which of the $s_o$ outputs of the network will correspond to which objects, so we must perform a matching step. The network produces distributions across possible object classes and bounding box sizes, so we try to best match the outputs to the ground truth information we have available. We construct a reference vector by concatenating one-hot encodings of the class and bounding box anchor for a given object. Then we compare these reference vectors to the output distributions produced at each slot. The Hungarian method is used to perform a maximum matching step such that ground truth annotations are assigned to the best possible slot, but no two annotations are assigned to the same slot.

Matching for relationships is similar. The ground truth reference vector is constructed by concatenating a one-hot encoding of its class with the output embeddings $h_s$ and $h_t$ from the source and destination vertices, $v_s$ and $v_t$. Once the best matching has been determined we have a correspondence between the network predictions and the set of ground truth annotations and can now apply the various losses. We also supervise the score for each slot depending on whether or not it is matched up to a ground truth detection - thus teaching the network to indicate a "full" or "empty" slot.

This matching process is only used during training. At test time, we extract object and relationship detections from the network by first thresholding on the heatmaps to find a set of candidate pixel locations, and then thresholding on individual slot scores to see which slots have produced detections.

## 4 Implementation details

We train a stacked hourglass architecture [21] in TensorFlow [1]. The input to the network is a 512x512 image, with an output resolution of 64x64. To prepare an input image we resize it is so that its largest dimension is of length 512, and center it by padding with zeros along the other dimension. During training, we augment this procedure with random translation and scaling making sure to update the ground truth annotations to ignore objects and relationships that may be cropped out. We make a slight modification to the orginal hourglass design: doubling the number of features to 512 at the two lowest resolutions of the hourglass. The output feature length $f$ is 256. All losses - classification, bounding box regression, associative embedding - are weighted equally throughout

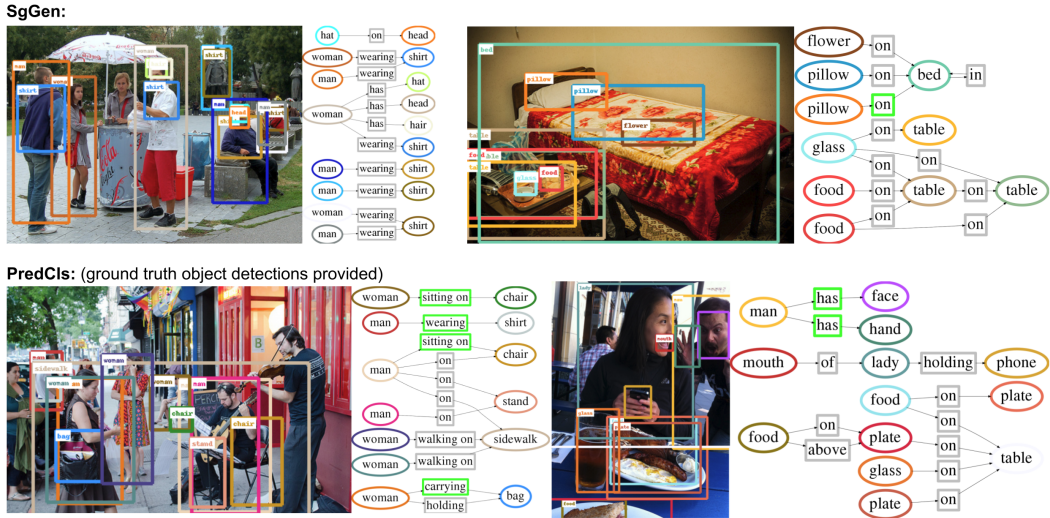

Figure 3: Predictions on Visual Genome. In the top row, the network must produce all object and relationship detections directly from the image. The second row includes examples from an easier version of the task where object detections are provided. Relationships outlined in green correspond to predictions that correctly matched to a ground truth annotation.

the course of training. We set $s_o = 3$ and $s_r = 6$ which is sufficient to completely accommodate the detection annotations for all but a small fraction of cases.

**Incorporating prior detections:** In some problem settings, a prior set of object detections may be made available either as ground truth annotations or as proposals from an independent system. It is good to have some way of incorporating these into the network. We do this by formatting an object detection as a two channel input where one channel consists of a one-hot activation at the center of the object bounding box and the other provides a binary mask of the box. Multiple boxes can be displayed on these two channels, with the first indicating the center of each box and the second, the union of their masks.

If provided with a large set of detections, this representation becomes too crowded so we either separate bounding boxes by object class, or if no class information is available, by bounding box anchors. To reduce computational cost this additional input is incorporated after several layers of convolution and pooling have been applied to the input image. For example, we set up this representation at the output resolution, 64x64, then apply several consecutive 1x1 convolutions to remap the detections to a feature tensor with $f$ channels. Then, we add this result to the first feature tensor produced by the hourglass network at the same resolution and number of channels.

**Sparse supervision:** It is important to note that it is almost impossible to exhaustively annotate images for scene graphs. A large number of possible relationships can be described between pairs of objects in a real-world scene. The network is likely to generate many reasonable predictions that are not covered in the ground truth. We want to reduce the penalty associated with these detections and encourage the network to produce as many detections as possible. There are a few properties of our training pipeline that are conducive to this.

For example, we do not need to supervise the entire heatmap for object and relationship detections. Instead, we apply a loss at the pixels we know correspond to positive detections, and then randomly sample some fraction from the rest of the image to serve as negatives. This balances the proportion of positive and negative samples, and reduces the chance of falsely penalizing unannotated detections.

## 5 Experiments

**Dataset:** We evaluate the performance of our method on the Visual Genome dataset [14]. Visual Genome consists of 108,077 images annotated with object detections and object-object relationships, and it serves as a challenging benchmark for scene graph generation on real world images. Some

| | SGGen (no RPN) | | SGGen (w/ RPN) | | SGCls | | PredCls | |
|---|---|---|---|---|---|---|---|---|
| | R@50 | R@100 | R@50 | R@100 | R@50 | R@100 | R@50 | R@100 |
| Lu et al. [18] | – | – | 0.3 | 0.5 | 11.8 | 14.1 | 27.9 | 35.0 |
| Xu et al. [26] | – | – | 3.4 | 4.2 | 21.7 | 24.4 | 44.8 | 53.0 |
| Our model | 6.7 | 7.8 | **9.7** | **11.3** | **26.5** | **30.0** | **68.0** | **75.2** |

Table 1: Results on Visual Genome

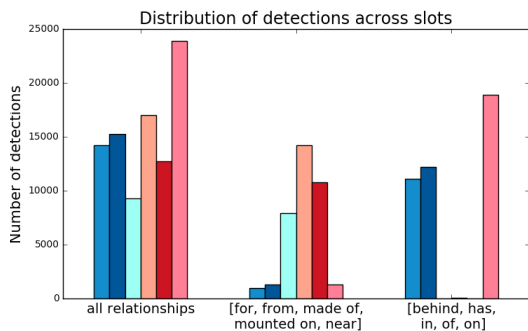

Figure 4: How detections are distributed across the six available slots for relationships.

| Predicate | R@100 | Predicate | R@100 |
|---|---|---|---|
| wearing | 87.3 | to | 5.5 |
| has | 80.4 | and | 5.4 |
| on | 79.3 | playing | 3.8 |
| wears | 77.1 | made of | 3.2 |
| of | 76.1 | painted on | 2.5 |
| riding | 74.1 | between | 2.3 |
| holding | 66.9 | against | 1.6 |
| in | 61.6 | flying in | 0.0 |
| sitting on | 58.4 | growing on | 0.0 |
| carrying | 56.1 | from | 0.0 |

Table 2: Performance per relationship predicate (top ten on left, bottom ten on right)

processing has to be done before using the dataset as objects and relationships are annotated with natural language not with discrete classes, and many redundant bounding box detections are provided for individual objects. To make a direct comparison to prior work we use the preprocessed version of the set made available by Xu et al. [26]. Their network is trained to predict the 150 most frequent object classes and 50 most frequent relationship predicates in the dataset. We use the same categories, as well as the same training and test split as defined by the authors.

**Task:** The scene graph task is defined as the production of a set of subject-predicate-object tuples. A proposed tuple is composed of two objects defined by their class and bounding box and the relationship between them. A tuple is correct if the object and relationship classes match those of a ground truth annotation and the two objects have at least a 0.5 IoU overlap with the corresponding ground truth objects. To avoid penalizing extra detections that may be correct but missing an annotation, the standard evaluation metric used for scene graphs is Recall@$k$ which measures the fraction of ground truth tuples to appear in a set of $k$ proposals. Following [26], we report performance on three problem settings:

**SGGen**: Detect and classify all objects and determine the relationships between them.

**SGCls**: Ground truth object boxes are provided, classify them and determine their relationships.

**PredCls**: Boxes and classes are provided for all objects, predict their relationships.

SGGen corresponds to the full scene graph task while PredCls allows us to focus exclusively on predicate classification. Example predictions on the SgGen and PredCls tasks are shown in Figure 3. It can be seen in Table 1 that on all three settings, we achieve a significant improvement in performance over prior work. It is worth noting that prior approaches to this problem require a set of object proposal boxes in order to produce their predictions. For the full scene graph task (SGGen) these detections are provided by a Region Proposal Network (RPN) [24]. We evaluate performance with and without the use of RPN boxes, and achieve promising results even without the use of proposal boxes - using nothing but the raw image as input. Furthermore, the network is trained from scratch, and does not rely on pretraining on other datasets.

**Discussion:** There are a few interesting results that emerge from our trained model. The network exhibits a number of biases in its predictions. For one, the vast majority of predicate predictions correspond to a small fraction of the 50 predicate classes. Relationships like "on" and "wearing" tend to completely dominate the network output, and this is in large part a function of the distribution of ground truth annotations of Visual Genome. There are several orders of magnitude more examples for

"on" than most other predicate classes. This discrepancy becomes especially apparent when looking at the performance per predicate class in Table 2. The poor results on the worst classes do not have much effect on final performance since there are so few instances of relationships labeled with those predicates.

We do some additional analysis to see how the network fills its "slots" for relationship detection. Remember, at a particular pixel the network produces a set of dectection and this is expressed by filling out a fixed set of available slots. There is no explicit mapping telling the network which slots it should put particular detections. From Figure 4, we see that the network learns to divide slots up such that they correspond to subsets of predicates. For example, any detection for the predicates *behind, has, in, of,* and *on* will exclusively fall into three of the six available slots. This pattern emerges for most classes, with the exception of *wearing/wears* where detections are distributed uniformly across all six slots.

# 6    Conclusion

The qualities of a graph that allow it to capture so much information about the semantic content of an image come at the cost of additional complexity for any system that wishes to predict them. We show how to supervise a network such that all of the reasoning about a graph can be abstracted away into a single network. The use of associative embeddings and unordered output slots offer the network the flexibility necessary to making training of this task possible. Our results on Visual Genome clearly demonstrate the effectiveness of our approach.

# 7    Acknowledgements

This publication is based upon work supported by the King Abdullah University of Science and Technology (KAUST) Office of Sponsored Research (OSR) under Award No. OSR-2015-CRG4-2639.

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
