[Reviews · NeurIPS 2017]

Reviewer 1



This paper proposes the use of a Hourglass net with associative embeddings to generate a graph (relating objects and their relationships) from an image. The model is presented as one of end-to-end learning. The Hourglass net provides heatmaps for objects and relationships, feature vectors are extracted from the top locations in the heatmaps and used with FC layers for predicting object-classes, bounding boxes and relationships among objects. Associate embeddings are used to link vertexes and edges, each vertex having an unique embedding. Experimental results show the high performance of the proposed methodology. I think this is a valuable work, yet I have doubts on whether it meets the NIPS standards. On the one hand, * the performance of this methodology is quite good in the approached subtasks. On the other hand, the novelty/originality of this work is somewhat limited: * it is based in a standard Hourglass network, standard in the sense that nothing novel seems to be added to the network, it simply estimates heatmaps and the FC layers provide estimates * it incorporates associative embeddings to generate the graph. Whereas this is an interesting approach, it is quite similar as the way it is used in [20] https://arxiv.org/pdf/1611.05424.pdf Hence, this work is another use for Hourglass-nets + associative embeddings as proposed in [20] obtaining good performance. I do not think this is enough contribution to be presented at NIPS. Further detailed comments: * The authors themselves acknowledge one of their contribution is a "November el use" of associative embeddings. Whereas I agree there is novelty on the usage of this methodology, the contribution in this side is limited (in the nips sense) * The authors fail at explaining both, the details and intuition of the proposed methodology (in all parts of the paper, but mostly in the intro, where this info is mostly important). The authors, assume the reader has the same degree of understanding of their proposed methodology ( not only of general technical stuff related to the approach, but also of methodology itself), making it kind of tedious to read the paper, and a bit complicated to follow it * Authors should make more explicit the differences and emphasize the importance of the contribution of this paper, certainly it is a different problem than that approached in [20], but is that difference enough contribution as to publish a paper in NIPS?, please argue on the differences and provide arguments supporting the strength of your contribution I removed a comment on overlap, my recommendation is not based on such assumption Typo * "them. [18]"

Reviewer 2



With this paper the authors evaluate the feasibility of a cnn-based approach to produce graphs of entities and relation between them directly from the raw image pixels. As for many papers following this kind of path, this study is a bit lacking in details and experimental evaluation, however it suggests an interesting approach.

Reviewer 3



Overall Summarization: This paper is an application of "Associative Embedding: End­-to-­End Learning for Joint Detection and Grouping"[20]. The author apply existing techniques from [20] to the task of detecting scene graphs from images. They first convert scene graph generation as a spatial graph detection problem, apply [20] to inference the spatial location of vertices as well as edges and group them together to form the outputs. The entire pipeline is well engineered and produces very impressive result. Despite the performance improvement, little technical novelty with regard to machine learning is introduced in the paper. Detailed Comments: - Section 3.2 Connecting elements with associative embedding * There is little justification about L_{push}. Why it is necessary to change this penalty term from the one described in [20]? Is there any empirical justification? - The writing and organization of paper needs improvement, a few places needs more explanations: 1) "We construct a reference vector … output distributions produced at each slot" (Line 227 - 228): How bounding box anchor is converted into a reference vector seems not clear. 2) "The input is 512x512 image …"(Line 240-241): What preprocessing steps is used here? Did you perform resize or crop or any other operations to the original image? 3) "If provided with a large set of detections…"(Line 251 - 254): How this additional input are incorporated after several layers of convolution and pooling is applied to the input image? Is there a sub-network or any kinds of fusion? 4) In figure 4, what does each slots mean in this figure?

Reviewer 4



Although I am totally no expert in this field, this looks a solid paper to me.